# Causal Reasoning and Event Cognition as Evolutionary Determinants of Language Structure

**DOI:** 10.3390/e23070843

**Published:** 2021-06-30

**Authors:** Peter Gärdenfors

**Affiliations:** 1Department of Philosophy and Cognitive Science, LUX, Lund University, Box 192, S-221 00 Lund, Sweden; Peter.Gardenfors@lucs.lu.se; 2Palaeo-Research Institute, University of Johannesburg, P.O. Box 524, Auckland Park ZA-2006, South Africa

**Keywords:** causal cognition, event representation, evolution of language, sentence structure, word classes

## Abstract

The aim of this article is to provide an evolutionarily grounded explanation of central aspects of the structure of language. It begins with an account of the evolution of human causal reasoning. A comparison between humans and non-human primates suggests that human causal cognition is based on reasoning about the underlying forces that are involved in events, while other primates hardly understand external forces. This is illustrated by an analysis of the causal cognition required for early hominin tool use. Second, the thinking concerning forces in causation is used to motivate a model of human event cognition. A mental representation of an event contains two vectors representing a cause as well as a result but also entities such as agents, patients, instruments and locations. The fundamental connection between event representations and language is that declarative sentences express events (or states). The event structure also explains why sentences are constituted of noun phrases and verb phrases. Finally, the components of the event representation show up in language, where causes and effects are expressed by verbs, agents and patients by nouns (modified by adjectives), locations by prepositions, etc. Thus, the evolution of the complexity of mental event representations also provides insight into the evolution of the structure of language.

## 1. Introduction

Human languages show a great deal of variation, but there are features that seem to be universal. In the Chomskyan tradition, such features are postulated to derive from an innate universal grammar. Within this tradition, however, evolutionary accounts of the universal grammar have focused on the role of recursion [1] (or, in Chomsky’s later writings, e.g., [2], the rule of ‘merge’), but Jackendoff [3] and Heine and Kuteva [4] have presented some more general principles. In this article, my aim is to provide answers to some fundamental questions concerning the structure of language on the basis of an account of the evolution of human causal cognition.

In regards to the structure of language, I focus on three questions:

(1) Why are sentences central units in language?

In all languages, sentences are fundamental units, but evolutionary linguists have hardly considered *why* this is so. My answer is, in brief, that sentences express *events*, and events are basic units of human causal thinking.

(2) Why do sentences consist of noun phrases and verb phrases? 

My answer to this question also derives from the structure of mental models of events. 

(3) Why are words in languages divided into classes? 

Again, I propose that common word classes can be explained by how we think about events.

The questions concerning the evolution of the structure of human language become more acute when comparing it with language-trained apes and other animals communicate. For example, bonobos produce sequences of signs without concern for whether they form sentences [5]. The complexity of human language is of a different order from what is found in non-human animal communication. The story of how the structure of language emerged in the hominin line is long, and it no doubt involved several steps.

The overarching question is, therefore, what happened to the human mind during our evolutionary past that can explain why our communication exhibits such structural complexity. In this article, my aim is to derive an answer to this question from an account of the evolution of human thinking about *causality*. My argumentation consists of three main steps: (i) A comparison between humans and non-human primates suggesting that human causal cognition is based on the reasoning about the underlying *forces* that are involved in phenomena, while other primates hardly understand external forces. I illustrate this by an analysis of the causal cognition required for early hominin tool use (Section 2). (ii) As a further evolutionary step, humans often organise causal relations in terms of *events*. I put forward a model where a representation of an event contains two entities representing a cause as well as a result, but also entities such as agents, patients, instruments and locations (Section 3). Section 4 is devoted to showing how communication about events yields evolutionary benefits. In particular, it opens up for cooperation about future goals. (iii) The components of the event representation show up in the structure of language, which will make it possible to answer the three questions above. Sentences form natural units since they refer to events (or states). In regards to word classes, causes and effects are expressed by verbs, agents and patients by nouns (modified by adjectives), locations by prepositions, etc. (Section 5). In conclusion, the evolution of the complexity of mental event components also provides clues to the evolution of the complex structure of language (Section 6).

## 2. What Is Special about Human Causal Cognition?

### 2.1. Non-Human Primate Reasoning about Causes

As background to the evolution of causal cognition in hominins, I present a summary of some of the findings concerning the abilities of non-human primates to reason with causes. This area goes back to Köhler [6], who, in his ground breaking experiments on chimpanzee planning, observed that apes had great difficulties in stacking boxes in order to be able to reach a banana that was hanging from the ceiling. He observes that when Sultan, the most cognitively proficient chimpanzee, tried to put a second box on top of a first, instead of doing so “as might seem obvious, began to gesticulate with it, … he put it beside the first, then in the air diagonally above, and so forth”. Similar observations of other apes, led Köhler to the conclusion that “there is practically no statics to be noted in the chimpanzee” ([6] (p. 149) and [7,8]).

When comparing causal cognition in non-human primates and in the hominin clade, it is useful to distinguish between *cued* (externally signalled) and *detached* mental representations [6,9]. A cued representation refers to something in the current (or recently experienced) external situation of the experiencer. When, for example, a chimpanzee uses two stones to crack nuts, it represents one of them as support (anvil) and the other as the hammerstone. By contrast, detached representations stand for objects or events that are not present in the subject’s current or recent external context and thus cannot trigger the representation directly. (This notion of detachment is related to Hockett’s [10] “displacement”, which is one of the criteria he uses to characterise what constitutes a language, but his criterion has a behaviourist touch to it).

An example is that when a chimpanzee plans to fish for termites with, it walks away and breaks off a thin stick from a tree, preparing it to fish with [11]. Forming mental images of the manufactured stick and how it is to be used are detached representations that become part of the chimpanzee’s plan.

An individual that has detached representations can create an inner world where consequences of different actions or events can be simulated [12,13,14]. Spreng and Grady ([15], p. 1112) argue that “remembering one’s past, imagining one’s future, and imagining the thoughts and feelings of others … are similar in that they all involve simulating an experience that is distinct from stimulus-driven behaviour”. Such simulations are central to abstract causal reasoning.

Being able to reason from inanimate effects to non-present causes seems, at present, to be unique to humans. There is a plethora of experiments and observations that indicate that primates cannot infer physical causes from their effects (e.g., [16,17]). For example, Cheney and Seyfarth’s [16] experiments with vervet monkeys showed that when catching sight of a predator, they emit warning cries. However, the same monkeys do not react to detached visual signs such as the trail of a snake or the carcass of an antelope in a tree, which indicates a leopard in the vicinity. Thus, while non-human primates are dependent on direct physical effects, it seems that the aptitude for causal understanding based on inanimate or indirect sensory cues evolved only in the hominin species [18,19,20].

The claim that reasoning from effects to non-present causes is unique to humans has been contended by other researchers. For example, in a study by Völter and Call [21] it was shown that chimpanzees, bonobos and gorillas can follow a trail left by a leaking yoghurt cup that was placed out of their sight to locate the yoghurt. A limitation of the result is that the apes did not use the trail when it did not match the food that was displaced—that is, when it could not be linked directly through sight, smell or taste.

Additional support for the thesis that non-human primates cannot reason about causes that are not perceived comes from an experiment by Civelek et al. [22] that examined how children and chimpanzees reasoned about unseen causes. The subjects saw a reward being dropped through an opaque forked tube into one of two cups. A sound signalled in which of the cups the reward was to be found. In one condition, the sound followed the dropping event, indicating that the cue was caused by the reward falling into the cup, and in another condition, the sound preceded the dropping event. Four-year-old children performed better in the first condition than in the second, which suggests that they understood the unseen cause. Chimpanzees and three-year-old children, however, performed at chance in both conditions.

Furthermore, Povinelli [17] presented some experiments which also indicate that chimpanzees (and other primates) have problems reasoning about the effects of gravitation on objects. These experiments have led to further investigations [23,24,25,26], and there is an ongoing debate (see [27,28]). Povinelli and Penn ([29], p. 77) conclude that “only humans are capable of second-order relational reasoning, and only humans, therefore, have the cognitive machinery that can support higher-order, theory-like, causal relations”. Johnson-Frey ([30], p. 201) also writes: “Comparative studies of chimpanzee tool use indicate that critical differences are likely to be found in mechanisms involved in causal reasoning rather than those implementing sensorimotor transformations”.

In a study comparing the nut-cracking performances of humans and chimpanzees [31], the result was that humans understood how to apply force to extract numerous nut species through using hammerstones. In contrast, chimpanzees only applied hammerstones to Panda nuts, although they regularly eat hard Irvingia nuts using their teeth. Chimpanzees in other groups and regions cracked different nut types with hammerstones [32,33], but a single group do not use hammerstones to obtain several food sources. This example illustrates how humans, in contrast to chimpanzees, reason more abstractly about the causal effects of applying tool-assisted forces. This allows humans to generalize a particular solution to a wider range of problems. 

Non-human animals can reason about an event involving actions—their own or those of others. The important step in reaching the more general event representation presented in the following section comes when not only actions but also other types of forces can function as causes in events. The ability to reason causally about detached forces—and not just actions, the human mind has evolved an extended capacity to reason and to plan that surpasses that of other primate species.

### 2.2. Human Reasoning about Forces

The human capacity to reason about of physical forces develops early in infants. Michotte [34] showed that if one object moving on a screen came into contact with another object and the other object began moving in the same direction, then adults perceived a causal relation between the two movements. If the second object only began moving 500 milliseconds after the collision, however, the time difference eliminated the impression of causality. Michotte’s experiments were performed with six-month-old infants by Leslie and Keeble [35]. The result was that the infants reacted differently to the two types of events. Leslie [36] argues that infants have a special system in their brains for mapping the “forces” of objects. Wolff and his collaborators [37,38,39,40,41] have collected further evidence supporting that people can directly perceive the forces that lie behind different kinds of events. The upshot is that the sensory input generated by the movements of an object is sufficient for the brain to automatically calculate the forces that lead to the movements [42].

Adults can also combine physical causes in their reasoning, as shown by Wolff [37]. The study showed that they can estimate the combined forces of a boat motor and the wind and to determine how the boat crosses a lake. Göksun et al. [43] extended this to a study of 3–5-year-olds. In addition to one-force events, the children were asked to predict the path of a ball that was influenced by two forces that were combined to reflect force dynamics patterns of “cause”, “enable” and “prevent”. The result was that while children could successfully reason about the one-force events, they had problems with a second force, incorporating it only if the two forces move in the same direction. The older the children, the more successful they were in accounting for the effects of the second force [44]. These experiments suggest that human abstraction and reasoning about physical forces develop over age, although the general system for perceiving forces as causes is present already at an early age.

As humans, we do not only reason about physical forces but also about how *psychological* and *social* factors influence us. The increasing complexity of hominin societies has generated a highly developed “theory of mind”, that is, an understanding of how our emotions, desires, intentions and beliefs lead to different kinds of interactions between people [13,45,46,47,48]. By observing the actions of ourselves and others and through various processes of social learning, we infer the state of mind of other humans under the hypothesis that their theory of mind is similar to our own.

In such cases, we do not perceive the cause of another’s actions physically, but use our understanding of their inner state as a causal variable for their behaviours. This involves a separation of perceptual similarities from the causal ones that are determined from emotions, desires, intentions and beliefs. Thus, the mental entities form a class of hidden variables that act as social forces and which we add to our perception in order to understand causal relations. A theory of mind is, therefore an important extension of human causal cognition [49].

### 2.3. Tool Manufacture and Use Were Selective Factors for Reasoning about Forces

From an evolutionary perspective, the central question becomes: Why did only hominins evolve causal thinking that is based on forces? Gärdenfors and Lombard [50] argue that tool manufacture and use were contributing factors to advanced forms of causal reasoning. The key to the argument is that tools extend the potential of the hominins to act across space and time.

As regards spatial cognition, the visual field of primates is divided into peri-personal and extra-personal space. The peri-personal space (the region within reach around the body), makes it possible for an individual to see its field of action. Tool use extends the peri-personal space [51,52]). Even further extensions of the peri-personal space are achieved when the tool leaves direct control of the body and exerts its force at a distance—entering extra-personal space, which may have stimulated the development of causal reasoning about external forces [50]. For example, throwing an object, as many primates do, may be the first method of exerting a force at a distance (see [53] for a speculative account). Chimpanzees, however, do not hunt by throwing. When they hunt bushbabies with sharpened sticks, they thrust with direct force [54]. Apes do not have the accuracy, force and speed that make human throwing so effective and so dangerous [55].

Sometime during hominin evolution, the display function of throwing in apes was replaced by a physical function with the aim to hurt or kill prey or opponents. A thrower that could reason about the physical effects at a distance of a stone or a spear would be more successful in hunting or warfare than an individual that could not [56]. The transition from actions to forces as causes is a critical step in the evolution of causal reasoning that can be traced through the use of hunting tools and weapons. The mapping between cause and effect must be inferred from the consequences of the throw (for example, from the behaviour of an animal or enemy that is hit). From the archaeological record, it would seem that *Homo neanderthalensis* shared the capacity for such reasoning with early *Homo sapiens* [57,58,59]. *Homo erectus*, on the other hand, may have been limited to thrusting weapons [60,61], wherein causal understanding was identified through the stabber’s own bodily action.

An even more complex form of causal force is the use of poisoned arrows that operates for an extended period of time, sometimes across a long distance, and often out of the sight of the hunter (see [50,62]. When preparing and using a poisoned arrow, the hunter must rely on advanced forms of abstract causal reasoning and long-term planning.

To sum up this section, non-human animals understand causation only in terms of agency, while humans can also reason about causes in a detached way via forces that operate across space (action at a distance) and time. Among the forces, one also finds the mental variables that are involved in an extended theory of mind.

## 3. Event Cognition

### 3.1. A Cognitive Model of Events

In this section, I turn to the relation between causal thinking and event cognition. A considerable part of human cognition depends on representations of events [63,64,65,66,67]. We use events in causal reasoning, planning and communication. As I argue below, our episodic memory also depends on event structures [66].

A central feature of events is that they are based on causal relations: An event typically contains information about an agent who is the cause of an action that leads to a result related to a patient. Although event representations generally contain an agent, some do not involve any, for example, events of raining, falling, drowning, dying and growing. A representation of an event may also contain other “thematic roles” such as instrument, recipient and beneficiary [68,69]. Agents and patients are object categories with different properties. It is assumed that an agent is able to act, which in the proposed framework amounts to exerting forces. The core idea of the event model presented in [63] and [65] is that an event contains two vectors—the force of an action that drives the event, and the result of the force (see Figure 1) (More formal details of the model can be found in [63,64,65]).

An action is modelled as a force vector (or a pattern of force vectors as in running). The result of an event is modelled as a change vector representing the change of properties of the patient [63,67]. For example, when Oscar (the agent) pushes (the force vector) a table (the patient), the force exerted makes the table move (the result vector). Or, when Victoria boils the carrots, the result is that the carrots become soft. When the result vector is just a point (a null vector), that is, when the result is no change, then the event is a *state*. An important feature of the event model is that it captures a basic sense of causation: The action of the agent causes the change in the patient. The distinction between forces and changes of states [37,39], results in the fundamental division between causes and effects. A special case of the event model, expressed linguistically by intransitive constructions such as “Victoria is walking” and “Oscar is jumping”, is when the patient is identical with the agent. In this case, the agent exerts a force on itself. In other words, the agent modifies its own position or property in some domain of agent space (=patient space).

In philosophy and psychology, the causal relationship between the action and the effect is typically analysed as being between two events (see [70,71]). In contrast, the proposed model describes causation as a relation within an event. Furthermore, in contrast to traditional philosophical theories, the distinction between forces and changes of states also entails that the cause and the result are represented as two different kinds of entities.

To be sure, humans also reason about another form of causation, that is, *generic* causal relations, for example, “eating toadstools will make you sick” and “lions kill people” [72]. Such generics concern relations between concepts and not causation within single events. There exist many attempts within economics and philosophy to reduce this kind of causal relation to information theory in terms of the probabilities involved (e.g., [73,74]). However, in the case of “lions kill people” the probability of a lion killing someone is much lower than many other causes of death, so the content of the generic is very low if measured in terms of general information or entropy. Gärdenfors and Osta Vélez [72] argue that the strength of the generic depends on the frequency of the characteristic category that is in focus relative to a contrast class. In the case of “lions kill people”, this would mean that the proportion of lions that kill people is higher than the corresponding proportion for most other animals. The relevant contrast class is, however, context-dependent so that transforming this into an information-theoretic measure would involve several factors. I will return to the role of generics in Section 5.1.

Another aspect is that the forces are not the only components involved in human causal reasoning, but *counterforces* (forces exerted by the patient or contextual forces such as gravitation) are also accounted for. This aspect was pioneered in Talmy’s [75] “force dynamics” and is further developed in Wolff’s [37,38,39] dynamics model. Depending on how the “affector” force vector (produced by an agent) is related to a “patient” force vector to generate a result vector, subjects judge that the affector force either *causes*, *enables* or *prevents* an effect. These results indicate that subjects make a distinction between different kinds of causal relations. Talmy’s force dynamics is grounded in physical events, but it can also be used to represent psychological or social forces where components of a theory of mind are exploited.

### 3.2. Event Cognition, Planning and Episodic Memory

A central question is what have been the evolutionary selective factors that resulted in the extended human dependence on mental representations of events. A main part of the answer is that detached event cognition allows us to speculate about potential outcomes of actions, test and re-adjust our imaginative hypotheses, and shift attention from one target to another. It thereby allows generalisation by comparing the force and result in one event with those of another [64]. 

In particular, different forms of *planning* involve event cognition. A plan consists of a series of imagined actions as causes together with the expected effects of the actions. For example, a hunter imagines a series of events, some related to the previous movements of an animal, some as part of a plan to kill or catch it. An anthropological example is that when hunting with poisoned arrows, Kalahari San engage in “speculative tracking”, using working hypotheses gained from the signs left by an animal, socially and experientially gained knowledge about the behaviours of the animal and of the landscape in which the tracking is taking place [76]. This imagination may also involve an understanding of the mental state of the animal, for example, that the animal is overheated or dehydrated. Based on these imagined reconstructions, the hunter creates predictions in ever changing circumstances involving a continuous cognitive process [77]. The upshot is that event cognition allows for ever more complex causal thinking to evolve.

One must, however, distinguish between immediate planning for present goals and *prospective* planning for future goals (Gulz [78] calls prospective planning anticipatory planning, a term that was also used in [13,79]). The crucial distinction is that for an individual to be capable of prospective planning, it must have a detached representation of its future goals. In contrast, immediate planning only depends on the current goals.

Several researchers have argued that the prospective skill for planning for future needs is exclusive to humans (e.g., [6,80,81,82]). This has been called the Bischof–Köhler hypothesis. In the light of recent experimental results, the hypothesis can, however, no longer be upheld. Great apes are not only able to select tools for future use [83,84], but also to save tools that have currently been used to obtain future goals [85]. The studies strongly suggest that great apes are able to outcompete current drives in favour of future ones [84,86] (This ability to plan for future needs also seems to have evolved independently in the avian taxon of corvids [87,88]).

My hypothesis that humans have more sophisticated cognitive representations of events than other species fits well with the theory that humans have an advanced *episodic memory*, allowing us to remember single events and the order in which they occurred [89,90].

The main neural correlate for both prospective planning and episodic memory is the hippocampal complex (hippocampus together with entorhinal cortex). Brain imaging shows that this complex is active in humans, both when they recall past events and when they imagine future ones [91]. The hippocampus can map out planned spatial paths, but also paths in other conceptual spaces [92,93]. This connection may be part of the explanation of why the hippocampus is central to both navigation and episodic memory—two seemingly unrelated cognitive capacities.

Related to episodic memory, the event model can handle *what-if* questions, that is, counterfactual reasoning concerning what would have happened if an action would have been different [94]. For example: “If I had hit the flint core less hard, it would not have broken”. Counterfactual reasoning seems to develop fully only relatively late in childhood. To wit, Rafetseder et al. [95] found that such reasoning is not fully developed in all children before 12 years of age, some of whom still lack an understanding of events that are causally dependent on counterfactual assumptions. According to Markovits’ [96] developmental patterns of conditional reasoning, fully developed counterfactual causal understanding is only reached between the ages of 14 and 16 (also see [97] on adolescent brain development in relation to counterfactual reasoning).

Apart from counterfactual reasoning, humans can also reason in terms of *omissive* causation, which concerns events that do not occur. For example, the fact that a hunter did not bring his spear caused him to be attacked by a lion. For many other models of causation, it is difficult to explain omissive causation, but the event model can also handle this (see [41,75,98]).

## 4. How Event Cognition Improves Communication

After these presentations of causal cognition and its role in mental representations of events, it is time to discuss the role of these systems in the evolution of language. Before I turn to how they influence the basic structure of language, this section is devoted to a presentation of what I view as the main evolutionary benefits of event representations in communication.

First of all, there are several levels of communication that are important to distinguish in an evolutionary setting. They will be introduced in Section 4.1. Next, symbolic words allow communication about things that are not present in the context. This will be the topic of Section 4.2. Many types of communicative tasks can be completed by using single words or a combination of a few words (or iconic signs). There are, however, two types of communicative situations, both unique to humans, where more complex linguistics structures are employed, namely cooperation for future goals (Section 4.3) and narratives (Section 4.4).

### 4.1. Levels of Communication

It is useful to distinguish three levels of communication, in addition to a basic level of cooperation. 

Level 0: *Praxis*. On this level, individuals interact with each other without using intentional communication.

If everybody in a cooperating group performs their assignments as expected, there is no need for explicit communication.

Level 1: *Evaluation and instruction*. On this level, valuations of the actions of others and of objects can be expressed by non-verbal approval or disapproval. The coordination of actions is achieved by instruction, expressed by wishes or demands. Linguistically, this is typically achieved by imperatives.

This form of communication is found in non-human animals and is among the first to develop in human infants.

Level 2: *Coordination of common ground*. On this level, individuals inform each other in order to be better coordinated. This is typically achieved via declaratives, but also questions can be used. 

On this level communication in the form of sentences becomes central.

Level 3: *Coordination of meanings*. On this highest level, individuals negotiate the meanings of words (signs).

This is the most advanced level, which assumes a meta-awareness of how language functions.

In the following, level 2 will be in focus since this is where human communication separates from that of other species.

### 4.2. Referring to Absent Objects

If the goal of collaboration is present in the current environment, for example, food to be gathered or an enemy to be fended off, the individuals need not communicate before acting. In contrast, if the goal is distant in time or space, then a shared representation of the goal must be obtained before cooperation can take place. For example, constructing a common dwelling involves coordinated planning of how to procure the building material and how to collaborate in the construction. Level 2 is therefore essential for cooperation about future goals.

A decisive difference between a symbolic language and the signals employed by animals is that signals only refer to what is present in the environment of the animal. For example, vervets only give warning calls when danger is immediate. *Signals* are about the surrounding world, while *symbols* often refer to our inner worlds, that is, to our imaginations, memories, plans and dreams. With the aid of symbols, humans can communicate about things that are not here and now or that may not even exist.

The use of symbols replaces cues from the environment in communication. If somebody has an idea about a common goal, she can use symbolic language to convey her thoughts. Furthermore, symbols make it possible for us to coordinate our knowledge, thereby creating a “common ground” [99] that can be exploited to create new forms of cooperation.

The possibility of referring to detached entities makes new forms of cooperation possible. In this way, a communicative system that makes it possible for members of a group to share mental representations of non-present entities becomes selectively advantageous [100,101].

To some extent, referring to objects that are not present on the scene can be done by just pointing [102]. For example, prelinguistic children about 12 months old can sometimes refer metonymically to an absent person by pointing to a place where that person has recently been or is normally located [103]. Chimpanzees can refer to absent objects by pointing, but only to invisible objects they know to be present [104]. They also point imperatively (level 1), while infants point declaratively (level 2) [105]. Pointing gestures are also frequently reintroduced in storytelling. Their function then is to give a visual complement to what the words point to in mental space [106,107] and [108] (p. 40) calls this “deixis at phantasma”. In this way, pointing begins to allow a “meeting of minds” [109].

Transiting from an animal signalling system to a fully symbolic language requires several steps. Bickerton [110] and other researchers [111,112,113], argue that during the evolution of language there was a stage when a protolanguage, which contained only the semantic components of language, was used. These authors, claim that *Homo erectus* mastered a protolanguage, and it only when *Homo sapiens* emerged on the scene when a language with a fully grammatical structure evolved. Everett [112] distinguishes between three levels of grammaticality (see [3] for another account of intermediary steps of grammatical evolution). The coordination of the common ground required for the forming of a common plan for future actions can possibly be achieved in a communication system that lacks syntax, that is, in a protolanguage. Nevertheless, as I argue in Section 5, some grammatical structure is needed in order to solve problem related to role assignments.

### 4.3. Communication for Future Cooperation

There exist many forms of cooperation in animals, for example, cooperative hunting, but the planning involved is individual, bound to the current context, and is not communicated intentionally. Of course, animals are reading the behaviour and maybe even the intentions of other individuals and use that to determine their own behaviour. This can be seen as a form of communication that improves cooperation, but it is not intentional.

When the planning involves several individuals, then variables of a theory of mind, such as the desires and intentions of other individuals, will also be part of the plan. For example, planning for collaborative hunting typically involves adjusting the plan to the presumed intentions of the other participants.

In contrast, planning for future collaboration seems to be more or less unique to the hominin line [100,114]. It essentially involves coordinating goals, which presumes several forms of coordination of common ground: coordination where something will be done (often outside the present visual field), joint reference to absent objects, coordination of goals and coordination of actions. This coordination is improved if the collaborators structure the plans in terms of future events. Thus, the evolution of event cognition made more advanced forms of collaboration possible.

Such planning requires that joint intentions are formed, which is an advanced form of the theory of mind, presumably unique to humans [13,45,48,115]. A *joint plan* can be analysed as a combination of forming a joint intention and coordinating actions. In previous work [13,45,79,101,116,117]. I have argued that symbolic language makes efficient cooperation about future goals possible. Along the same lines, Tylén et al. [118] (p. 6) write: 

Analogous to the way that manual tool use has been shown to enlarge the peripersonal space by extending the bodily action potential of arm and hand in space …, linguistic symbols liberate human interactions from the temporal and spatial immediacy of face-to-face and bodily coordination and thus radically expand the *interaction space.*

During the evolution that led to *Homo sapiens*, our hominin ancestors developed new forms of cooperation that made it possible to organise their societies in new ways. It is generally agreed that hominins evolved in open landscapes in which the individual travelled over long ranges [119]. Cooperative foraging could have been caused by increased seasonality and variability in the environments.

The Oldowan culture—the first along the Homo lineage—was signified by an extension in time and space [120]. There is clear evidence that the transport of artefacts (at least stone tools) was an important trait of the culture [121]. Isaac [120] also speculates that division of labour was present in the Oldowan culture. Such arrangements presume coordination of activities, which is an indication of detached communication. The more complex a culture is, the more effort must be made to preserve the complexity. Donald [122] writes: “Memory for a variety of special skills usually involves some division of labour, as well as a collaborative strategy for passing those skill on to every new generation”. Therefore, *teaching* about the environment and different procedures for food procuration became increasingly important [123,124].

Division of labour involves *role taking*. Spontaneous role taking can occasionally be found in social hunting in some non-human species, for example, chimpanzees [125]. If the role taking is flexible, however, so that an individual, for example, sometimes drives the game, sometimes acts as a lookout or sometimes waits in ambush, then each participant must have a representation of the roles of the other team members [126,127]. This also involves joint intentionality about the activities of the team [47]. Furthermore, if the role taking is part of a plan, then the roles must be communicated in some way. As I argue in Section 5.2, communication about role assignments maps onto fundamental syntactic principles. 

The analysis presented here dovetails with Smith [128] (p. 241), who argues that linguistic communication has the following benefits for cooperation:(1)Simplifies otherwise difficult coordination problems, especially those involving many agents and planning for future events;(2)Reduces the cost of enforcing adherence to collectively beneficial norms;(3)Enhances the efficiency of signals, including those which provide collective goods;(4)Facilitates the positive assortment of individuals who adhere to similar norms and conventions.

### 4.4. Who Did What to Whom? The Adaptive Benefits of Gossip

In social species, individuals often must decide whether to cooperate or not. Within game theory, in the investigations of prisoners’ dilemmas and similar conflict situations, it is taken for granted that the players know who the potential collaborators are. In practice, however, the most important question is: How do you know *whom* to cooperate with? Dessalles [111] (p. 360) writes: “Some of our ancestors who belonged to the first species of Homo, say, began to form sizeable coalitions. In such a “political” context, finding good allies becomes essential”. Sharing information about the other members of the group is an important special case of coordinating common knowledge.

Reciprocal altruism (“you scratch my back, and I’ll scratch yours”) is found in several animal species. Such cooperation presumes trust between two individuals. *Indirect*
*reciprocity* is a more extreme form of altruism: “I help you, and somebody else will help me”. This form of cooperation involves a group of individuals. The conditions for indirect reciprocity to evolve as an evolutionarily stable strategy have been investigated [129,130]. The key concept in Nowak and Sigmund’s [129] evolutionary model is that of the *reputation* of an individual: An individual X’s reputation is built up from how members of the society observe X’s behaviour towards third parties and then how this information is spread to other members of the society. The higher the reputation of X, the more other individuals are willing to cooperate with X.

The communication required for functioning forms of indirect reciprocity typically concerns whom you can trust. The information is generally conveyed in the absence of the individual who is evaluated. This process makes *gossip* becomes an important way of achieving consensus about reputation [131,132]. In this way, the information that X is a “selfless” helper can be shared knowledge within the group. T X’s reputation can then be evaluated by any individual who needs to decide whether or not to assist X in a troublesome situation. Evidently, reputation cannot be directly observed by others in the same way as such status markers as a raised tail among wolves. In contrast, each individual must keep an account of the reputation of the other individuals with whom she considers interacting. Semmann et al. [133] experimentally demonstrate that building a reputation through cooperation is valuable for future social interactions, not only within but also beyond one’s social group. Gossip, therefore, plays a central role in the evolution of language.

Gossip typically contains expressions of the form “X did A to Y”, which require thematic roles such as agent, action and patient to be identified. A communication system that communicates this form of messages must be able to (a) refer to individuals in their absence, for example, by names, (b) express that “X was good to Y” and “Y was bad to X” and (c) to keep track of the roles of X and Y. In order to evaluate X and Y as potential collaborators, such expressions are clear examples of ascribing roles to persons. They are also difficult to convey without ambiguity within a protolanguage: without markers for roles, since “X was good to Y” can then not be distinguished from “Y was good to X”. Such a system must thus contain some forms of syntax, maybe of the first type described by Everett [112].

Providing information about whom to cooperate with is a central function of gossip, but not the only one. Dunbar [131] argues that it functions as a replacement for grooming and thus results in stronger bonding between individuals. Furthermore, other forms of information about individuals not related to their trustworthiness can also be important in the complex net of strategic interactions between the members of a group.

I have now presented two forms of communication for cooperation where sentences are required: coordination of future goals and gossip. Both types typically take the form of narratives. Pragmatically, their function is to coordinate the common knowledge of the interlocutors (level 2). I am not claiming that communication on level 2 can only be achieved by sentences. I have already pointed out that pointing and other forms of gestures can also perform this function (see also [134]), and maybe even drawing could achieve this. Nevertheless, sentential structures strongly amplify the efficiency of communication on level 2. 

What is important is that describing planned actions as well as information about relations between different individuals are special cases of describing events. My hypothesis is that the capacity to communicate about events is a crucial distinction between the communication of language-trained apes and that of humans.

## 5. From Events to Sentences

In this section, I show how causal cognition and the structure of event representations provide answers to the three questions concerning the fundamental structure of language that were presented in the introduction.

### 5.1. Sentences Are Needed for the Coordination of Common Knowledge

There is one linguistic unit that is central to both types of communication discussed in the previous section: the *sentence*. The first question on my list concerns *why* we organise much of our communication in sentences. The Chomskyan tradition takes it for granted that the goal of speaking is to generate sentences, having a minimal structure of a noun phrase and a verb phrase. Within this tradition, the main problem is to decide whether certain combinations of words are grammatical—the meaning of a sentence is secondary. Furthermore, in cognitive linguistics, sentences are seen as natural units [69,135,136,137,138].

In analytic philosophy, sentences are also central units. In the tradition since Frege, the content of a sentence is a proposition. As an answer to my first question, Frege writes that the role of sentences is to express *thoughts*. This answer is simply not sufficient unless it can be determined how a thought is to be identified (independently of language).

If one takes a cognitive perspective on communication, an explanation of why we express ourselves in sentences is needed. My proposal is that the primary function of sentences is that they *refer to events*. I next outline how a mental representation of an event can be expressed in sentences describing different aspects of the event that are relevant to the communicative situation. Not only what the interlocutors perceive, but also what they are attending to, their current goals, their future plans and their previous communication will influence how sentences are formed.

It is important that I am not claiming that all human communication takes the form of sentences. First of all, from an evolutionary perspective, the focus should be on utterances rather than sentences. Utterances can only be evaluated in relation to a communicative context that contributes to their meaning. In contrast, within philosophy (and to a large extent also linguistics), it is assumed that sentences have a meaning that is independent of the context. Second, there several forms of communication, for example, evaluative and emotional expressions, requests and demands, which do not refer to events and which may not even require symbolic structures.

My focus in this article is, however, the communication required for advanced cooperation, and thus primarily for communication on level 2, that is, coordinating inner worlds. For this task, sentences prove their mettle. In present-day human communication, sentential structures are so entrenched in the linguistic structure that they are also used for communication on level 1, which is evaluative expressions and demands, and on level 3, which is, coordinating meanings of words and other meta-functions of language.

Events are complex phenomena, and they cannot be exhaustively described. Before a sentence can be generated, it must first be decided which information to include. A speaker has the liberty to express different perspectives on the same event. The choice of perspective determines what is expressed in a sentence. In cognitive linguistics, this choice is called the *construal* of the sentences. One manner of expressing different perspectives is via the choice of grammatical construction. For example, in “Victoria paints Oscar’s face”, Victoria is more in focus, while in the passive construction “Oscar’s face is painted by Victoria” the focus is on Oscar’s face. One must therefore distinguish two different levels of meaning: (i) the representation of an event and (ii) a construal that picks out certain aspects of the event.

How is a construal of an event determined? The *attention* of the speaker is a selection mechanism for a cognitive event representation, as seen in the example above. There are, however, other aspects of how a construal is formed (see [139] (Ch. 3) for a survey). One is *perspective*: For example, if you and I are standing on two sides of a house, I can say that you are behind the house if I put myself in the centre (egocentric view), or I can say that you are in front of the house if I put the house and the direction of its main side in focus (allocentric view). By analogy with the visual process, a construal focuses only on certain parts of an event. For example, the sentences “Oscar sprayed paint on the wall” and “Oscar sprayed the wall with paint” describe the same event with the aid of two different construals. The difference between them is that in the first, “paint” is focused on as the patient of the action, while in the second, “the wall” is made the patient [69] (p. 124). The attention process is then applied to select the force or the result vector of the event model, generating a construal that can be used in the language production model. For example, in “Oscar scrubs the floor”, the force vector is selected for the construal, while in “Oscar cleans the floor”, it is the result that is in focus.

On the basis of the notion of a construal of an event, I can now formulate a fundamental connection between the semantics of sentences (utterances) and events. Jackendoff [140] (p. 327) makes a similar proposal: “[T]he category corresponding to a sentence is an event or a state rather than a truth value”. However, his event model is algebraic rather than geometric, as proposed here).

*Thesis about sentences*: A (declarative) sentence expresses a construal of an event.

In linguistics, a tight mapping between linguistic expressions and construals of events is in general assumed [141]. DeLancey [142] provides good linguistic arguments why events must be distinguished from their construals.

The thesis is constrained to declarative sentences. First, on communication level 1 (evaluation and instruction), evaluative expressions or imperatives are typically used instead. Evaluative expressions need not even involve words but can consist of, for example, an affirming nod. Since an imperative relates to the attitudes of the speaker and is normally directed to the addressee, imperatives typically omit some event elements that occur in sentences (“Salt please”). Second, on level 2 (coordination of common ground), a common form of utterances are *generics*. They have a different function in communication in that they provide information about the underlying semantic structure. Gärdenfors and Osta Vélez [72] distinguish between two kinds: (a) Property generics dealing with characteristic properties of objects (”ducks lay eggs”), and (b) event generics dealing with causal relations within event types (“sharks kill people”). Finally, in a seeming violation of the thesis, there also exist declarative sentences used on communication level 2, such as the current one, where information is given without referring to an event. Such sentences, however, communicate abstract thoughts and thereby constitute advanced uses of language (and meta-language) that have evolved at later stages.

### 5.2. Noun Phrases and Verb Phrases

The second question concerns why sentences are built up from noun phrases and verb phrases. In Heine and Kuteva’s [4] “layers” of grammaticalisation, nouns constitute the first layer and verbs the second. Nouns and verbs are also the only word classes that are fairly stable cross-linguistically. Why are these classes so fundamental? 

I submit that the answer to this question also derives from event cognition to wit, the minimal construals of events. Agents and patients are linguistically expressed by nouns (including names) or noun phrases and force vectors and results vectors by verbs. 

Linguists distinguish between manner verbs and result verbs—where “manner verbs specify as part of their meaning a manner of carrying out an action, while result verbs specify the coming about of a result state” [143] (p. 21) (see also [138] (Ch. 1)). This distinction maps directly to the distinction between force vectors and result vectors in events. For example, “push” refers to the force vector of an event, “move” refers to changes in the spatial domain of the result vector and “heat” refers to changes in the temperature domain (Possible exceptions to this general rule are verbs such as “split” that describe changes in which objects exist and verbs such as “give” that describe intentional actions involving recipients. These are discussed in [63] (Section 10.3.2)).

There are four minimal combinations of elements from an event model that are expressed as sentences: (i) agent + force vector (“Oscars scrubs”), (ii) agent + result vector (“Oscar cleans”), (iii) patient + force vector (“the table is scrubbed”), and (iv) patient + result vector (the table is cleaned). Each of these combinations yields a complete sentence. This leads to the following thesis [63] (Ch. 11).

*Thesis about event construals:* A construal of an event contains at least one vector (force or result) and at least one object (patient or agent).

This thesis requires that at least an agent or a patient (expressed by a noun phrase) and a force vector or a result vector (expressed by a verb phrase) are parts of what is expressed in a sentence. The model thus explains the basic distinction between nouns phrases and verb phrases and why at least one of each typically occurs in a sentence. The thesis thereby provides motivation for why a construction consisting of a noun phrase and a verb phrase is a cognitive unit of communication.

However, the thesis is also supported by studies of grammaticalisation. Heine and Kuteva [4] (p. 119) write: “All evidence from grammaticalisation leads to the same hypothesis, namely that the earliest structure of human language was lexical in nature, first consisting only of noun-like utterances before verbal utterances appeared, thereby making it possible to form propositional constructions”.

In linguistics, events are often modelled using symbolic notation [140,144]. For example, Rappaport Hovav and Levin [144] (p.116) represent the meaning of the verb “break” as follows:[[X ACT_<MANNER>_] CAUSE [Become [Y <BROKEN>]]]
This can be rendered as “X acts in a manner to cause Y to become broken”. In this kind of analysis, however, the linguistic level is still present since the verb “break” reappears as <BROKEN>.

The ACT-CAUSE-BECOME model of verb semantics presented by Rappaport Hovav and Levin can, nevertheless, be mapped onto the present proposal. The ACT of the formalism corresponds to the force vector, except that not all force vectors involve action. The BECOME corresponds to the result vector (Goddard and Wierzbicka [145] argue that “happen to” fits better than “become”). CAUSE is the mapping from force vector to result vector [64]. As should be clear by now, the model presented here is richer since the force and result vectors are grounded in a theory of causal cognition. This connects the semantics of causation and events directly to bodily and perceptual variables.

### 5.3. Other Word Classes

The third question concerns why the words of a language can be grouped into a small number of classes. I have already motivated why there are nouns and verbs since they express the most central components of an event. The basic constituents of events —agent, patient, action and result—can, however, be augmented with other thematic roles such as recipient (“Oscar handed Victoria a towel”) and instrument (“Oscar painted the house with a roller”). Nouns and verbs can, however, be modified in many different ways. Heine and Kuteva [4] show how nouns and verbs have historically been transformed to other word classes. I will only briefly discuss adjectives and prepositions here. The semantic roles of these and other word classes are treated more extensively in the second part of [63].

An adjective has two major communicative functions. The first is as a *specification* of a noun (or noun phrase) that contributes to the identification of a referent. For example, if you want somebody to fetch you a cup and there are several cups present in the context, you specify it further by adding an adjective, saying “the blue cup” or “the large cup”.

The second function of an adjective is *informative*. You can say, “The stove is hot”, as a warning to somebody. In linguistic terms, the adjective is then a complement to a copula (“is”) or an intransitive verb (“the meal tastes wonderful”). The two functions of adjectives may very well be cognitively separated. Therefore, it is not necessary that these functions are expressed by one word class. In line with this, Dixon [146] (p. 30) notes that some languages have two different word classes, one fulfilling the specification function and another fulfilling the informative function. As a matter of fact, some words classified as adjectives in English only have one of the functions: “afraid” and “alive” can only be used informatively, and “absolute” and “main” can only be used as a specification [147].

Most prepositions can be grouped into two classes: *locative*, indicating where something is, and *directional*, indicating where something is going [148]. Locative prepositions complement a noun (noun phrase) by specifying the location (a region) in the visuospatial domain: “Put the plate in front of grandfather!” This function is required for instruction and is similar to the specification function of adjectives. Another communicative function is handled by directional prepositions. In a sentence such as “Oscar went to the river”, the phrase “to the river” has the same function as a result verb: it specifies the result vector of an event. It should be noted that in both functions, the preposition is combined with a noun (or a noun phrase).

In communicating a plan for a collaboration, it is clear that both the specification function of adjectives and the spatial information contained in prepositional constructions add precision to the message. Even though a plan communicated by only nouns and verbs may succeed, the communication typically gains in content by exploiting further word classes.

## 6. Conclusions

In this article, I have argued that the evolution of causal cognition and event representations provides clues to the evolution of the complex structure of language. A consequence of this approach is that the evolutionary processes that led to language were cognitive and social, not primarily linguistic. Donald [122] (p. 2) writes: “If languages are products of cognitive interactions in groups, this fact alone would demand a culture-first theory of language genesis. … Archaeological evidence suggests that human ancestors were *skilled* long before they were *articulate*. … Therefore the cognitive apparatus for refining skill must have existed in some form before languages could emerge from group interactions”. This position entails that most human cognitive functions had been chiselled out by evolution before language appeared on the hominin scene. Language would not have evolved without these cognitive capacities, in particular advanced causal cognition, having a rich theory of mind, representing events, having a memory system that includes episodic memory and representing future goals [13,45,49,100].

In summary, my main thesis concerning the origins of linguistic structure is that sentences are natural units in communication because human cooperation has benefitted evolutionarily (and still benefits) from communication about events. I have argued that a declarative sentence refers to an event (or a state as a special case). Non-human animals do not refer to events since neither is their causal cognition rich enough nor do they have the mental capacity to cooperate about future goals. Therefore, there has been no selection for communication with a sentential structure in non-human species.

## Figures and Tables

**Figure 1 entropy-23-00843-f001:**
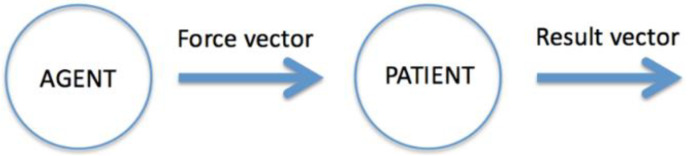
The main components of an event representation.

## Data Availability

Not Applicable.

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
