# Peer review of "Causal Reasoning and Event Cognition as Evolutionary Determinants of Language Structure"

_entropy, 2021, doi:10.3390/e23070843_

Round 1
Reviewer 1 Report
The description of the state-of-the-art is based on clear references. Research questions are clearly and explicitly stated. The conclusions are supported by the research data. The abstract is a concise description of the work. The introduction is well structured, and it covers all the concepts investigated in the methodological part. The previous work is well presented and integrated. The research design used is appropriate in order to answer the research questions proposed by the authors. The methods are described properly. The results are clearly presented and are in relation to the concepts investigated. The discussions are clear and concise. The conclusions are strongly related to the findings of the research work. All the format and style features were respected and are compliant with the requirements. Some current papers related to cognition should included, for example: Examination and comparison of the EEG based attention test with CPT and TOVA, The evaluation of BCI and PEBL-based attention tests or Examine the effect of different web-based media on human brain waves.
Author Response
The description of the state-of-the-art is based on clear references. Research questions are clearly and explicitly stated. The conclusions are supported by the research data. The abstract is a concise description of the work. The introduction is well structured, and it covers all the concepts investigated in the methodological part. The previous work is well presented and integrated. The research design used is appropriate in order to answer the research questions proposed by the authors. The methods are described properly. The results are clearly presented and are in relation to the concepts investigated. The discussions are clear and concise. The conclusions are strongly related to the findings of the research work. All the format and style features were respected and are compliant with the requirements. Some current papers related to cognition should included, for example: Examination and comparison of the EEG based attention test with CPT and TOVA, The evaluation of BCI and PEBL-based attention tests or Examine the effect of different web-based media on human brain waves.
Reply: Thank you for the positive report about the paper. I have checked the suggested references, but, unfortunately, they seem to be of little relevance for the topic of the paper.
Reviewer 2 Report
This is a very interesting article, which is logically structured and easy to follow. There are three research questions that are answered convincingly (within the scope of this paper). I definitely think that the paper deserves to be published.
That said, I still have some comments and questions:
Section 2.1: Köhler (1917) observed that apes had great difficulties stacking boxes. Is it clear that the apes were cooperative and were actually trying to stack boxes? What was the utility of this activity to the apes?
Section 2.3 line 244: "to for" (typo); there are a few more typos here and there throughout the manuscript; please proofread
Section 3.2: This section is about planning and episodic memory. It discusses to what extent animals and humans are able to plan and remember events. When reading this section I came to think of the book "Understanding Intelligence" by Pfeifer and Scheier (1999). A central claim by these authors is that there does not need to be intelligent intent behind behavior that seems intelligent to an observer. Similarly, what seems like smart planning, might not be smart planning. I believe that it could, for instance, be an expression on species-specific behavior. A hen scratching the ground and looking for seeds all the time might do so although she is not hungry at the moment. However, looking for food before you get hungry, as an instinct, might give evolutionary advantages. The manuscript contains
an example of great apes making tools and saving them for future use. This could demonstrate the capacity of planning for future events. However, it could also just be something that is fun and feels good to the apes and that is why they do it. Also humans do many things that they find fun or rewarding in themselves, which can be beneficial in the future, but that does not necessarily involve a component of planning for a goal. A child drawing pictures is not necessarily doing it in order to practice to become a professional artist.
Section 4 is interesting as a whole, although speculative. I agree that human language makes it possible to share knowledge about events that are not here and now: the event might in the past, planned for the future, or it might be a totally imaginary event. Language makes it possible to refer to absent objects and events using symbols and grammar. However, communication can also take place using drawing, for instance. Presenting a picture, or a series of pictures, which we find in ancient caves or medieval churches, tell us about past events or imagined events. Does the construal of events using language make use of different cognitive mechanisms than when using images?
The subsections 4.1 and 4.2 attempt to explain why sentence structure came to be what it is: 1) a means to communicate future, joint plans and 2) a means to narrate past events by "gossiping" and telling others about who did what to whom, this being essential in choosing allies. I am not fully convinced by the selection of these two particular explanations.
To start with the gossiping: if A is gossiping to B about what X did to Y, then according to the manuscript this is to help B know whether to collaborate with X, because B (who wasn't there) will know whether X did something good or bad. Another essential claim is that sentence structure is about construing events, so a sentence really needs to be able to convey the message of what X did to Y. My objections are the following: Firstly, if A is gossiping to B, it might primarily be about bonding between A and B, and A trying to push A's own reputation in the eyes of B (possibly, though, at the expense of X). Secondly, it is my impressions that the communities where the early hominids/humans lived were fairly small tribes, where everybody knew each other. Thus, B would not need to hear from A what X and Y are like. Preferences who to collaborate with and in which roles could well arise within the tribe using other means than language. Why this focus on the importance of gossip and questions about who did what to whom? Thirdly, language has other roles than relating facts about events. Cues could be found in language acquisition and the evolution of language skills in children. I believe that children and their parents communicate a lot about attitudes, feelings and needs: what I like (milk), what I dislike (water), what I need (food), what I do not want now (sleep), what is good, what is bad, etc. These attitudes can often be communicated using extra-linguistic means (crying, laughing, body language, etc), which could be learning signals in associating
the linguistic expression with its meaning. From this follows interactions, asking for things that are needed (give this, take that), where the subject (I) is present and is in the role of agent or patient (or recipient, experiencer etc).
The hypothesis of the manuscript is that language arises as a product of cognitive capabilities for planning and episodic memory combined with a need to coordinate plans and find allies. The nuclear sentence structure is supposed to be the construal of an event, where X does something to Y. Where do the attitudes of the speaker enter here? Does the speaker A think that X is doing something good or bad? Should the hearer B just make his or her own interpretation whether the action performed by X to Y was good or bad. Where is the central role of attitudes in sentence structure? Or is it not there? Additionally, is conveying an attitude a state rather than an event?
What about imaginary events? Could language be a device for imagination, taking pleasure in exploring events that are not here and now? Imagining things has been shown to improve actual performance; imagining is practising. What about talking about religion and supernatural forces? Religious beliefs are common in all cultures and apparently also in very early ones. Is it clear that these mentioned uses of language are secondary to the classical, and rather worn, example of the planning of cooperation during a hunt?
I am also very much speculating here, but I am not convinced that the construal of events is the all encompassing basis for sentence structure. What about sharing information about your inner state, what you need, what you want, what you feel towards something, requesting favors from others, telling others what to do, etc?
Author Response
This is a very interesting article, which is logically structured and easy to follow. There are three research questions that are answered convincingly (within the scope of this paper). I definitely think that the paper deserves to be published.
That said, I still have some comments and questions:
Section 2.1: Köhler (1917) observed that apes had great difficulties stacking boxes. Is it clear that the apes were cooperative and were actually trying to stack boxes? What was the utility of this activity to the apes?
Reply: A description of the goal of the activity has been added
Section 2.3 line 244: "to for" (typo); there are a few more typos here and there throughout the manuscript; please proofread.
Reply: This and other typos corrected.
Section 3.2: This section is about planning and episodic memory. It discusses to what extent animals and humans are able to plan and remember events. When reading this section I came to think of the book "Understanding Intelligence" by Pfeifer and Scheier (1999). A central claim by these authors is that there does not need to be intelligent intent behind behavior that seems intelligent to an observer. Similarly, what seems like smart planning, might not be smart planning. I believe that it could, for instance, be an expression on species-specific behavior. A hen scratching the ground and looking for seeds all the time might do so although she is not hungry at the moment. However, looking for food before you get hungry, as an instinct, might give evolutionary advantages. The manuscript contains
an example of great apes making tools and saving them for future use. This could demonstrate the capacity of planning for future events. However, it could also just be something that is fun and feels good to the apes and that is why they do it. Also humans do many things that they find fun or rewarding in themselves, which can be beneficial in the future, but that does not necessarily involve a component of planning for a goal. A child drawing pictures is not necessarily doing it in order to practice to become a professional artist.
Reply: Yes, it is true that there are many situations where animal behaviour seems like smart planning, for example squirrels collecting nuts for the winter, but where it can be shown that the behaviour is instinctive. However, in the experiments by Mulcahy and Call 2006, Osvath and Osvath 2008 and later ones, the experimental conditions exclude instinctive behaviours and really indicate prospective planning.
Section 4 is interesting as a whole, although speculative. I agree that human language makes it possible to share knowledge about events that are not here and now: the event might in the past, planned for the future, or it might be a totally imaginary event. Language makes it possible to refer to absent objects and events using symbols and grammar. However, communication can also take place using drawing, for instance. Presenting a picture, or a series of pictures, which we find in ancient caves or medieval churches, tell us about past events or imagined events. Does the construal of events using language make use of different cognitive mechanisms than when using images?
Reply: This is an interesting comment. It may very well be that early cave paintings represent events, but a question is what they are supposed to communicate. One speculation that has been put forward is that they were used in combination with a narrative. Of course, the cognitive mechanisms for using pictures is different from those exploited in linguistic communication. In any case, the possibility of pictorial communication does not conflict with, but rather complements, linguistic communication. And as far as is known linguistic communication far precedes pictorial in evolutionary time. I believe, however, it would go too far to discuss pictorial communication in the article.
The subsections 4.1 and 4.2 attempt to explain why sentence structure came to be what it is: 1) a means to communicate future, joint plans and 2) a means to narrate past events by "gossiping" and telling others about who did what to whom, this being essential in choosing allies. I am not fully convinced by the selection of these two particular explanations.
To start with the gossiping: if A is gossiping to B about what X did to Y, then according to the manuscript this is to help B know whether to collaborate with X, because B (who wasn't there) will know whether X did something good or bad. Another essential claim is that sentence structure is about construing events, so a sentence really needs to be able to convey the message of what X did to Y. My objections are the following: Firstly, if A is gossiping to B, it might primarily be about bonding between A and B, and A trying to push A's own reputation in the eyes of B (possibly, though, at the expense of X). Secondly, it is my impressions that the communities where the early hominids/humans lived were fairly small tribes, where everybody knew each other. Thus, B would not need to hear from A what X and Y are like. Preferences who to collaborate with and in which roles could well arise within the tribe using other means than language. Why this focus on the importance of gossip and questions about who did what to whom? Thirdly, language has other roles than relating facts about events. Cues could be found in language acquisition and the evolution of language skills in children. I believe that children and their parents communicate a lot about attitudes, feelings and needs: what I like (milk), what I dislike (water), what I need (food), what I do not want now (sleep), what is good, what is bad, etc. These attitudes can often be communicated using extra-linguistic means (crying, laughing, body language, etc), which could be learning signals in associating
the linguistic expression with its meaning. From this follows interactions, asking for things that are needed (give this, take that), where the subject (I) is present and is in the role of agent or patient (or recipient, experiencer etc).
Reply: I accept the reviewer’s comment and have modified the discussion of gossip in the text.
The hypothesis of the manuscript is that language arises as a product of cognitive capabilities for planning and episodic memory combined with a need to coordinate plans and find allies. The nuclear sentence structure is supposed to be the construal of an event, where X does something to Y. Where do the attitudes of the speaker enter here? Does the speaker A think that X is doing something good or bad? Should the hearer B just make his or her own interpretation whether the action performed by X to Y was good or bad. Where is the central role of attitudes in sentence structure? Or is it not there? Additionally, is conveying an attitude a state rather than an event?
Reply: Again, this is a good comment and I have added a new subsection 4.1 and some further comments on attitudes that I hope improve the arguments.
What about imaginary events? Could language be a device for imagination, taking pleasure in exploring events that are not here and now? Imagining things has been shown to improve actual performance; imagining is practising. What about talking about religion and supernatural forces? Religious beliefs are common in all cultures and apparently also in very early ones. Is it clear that these mentioned uses of language are secondary to the classical, and rather worn, example of the planning of cooperation during a hunt?
Reply: Of course, imaginary events are important. Already in creating a joint plan for future cooperation, imaginary events are involved. This capacity is then exploited (or exapted) in narratives and myths. In my opinion, however, these forms of communication are evolutionarily younger,
I am also very much speculating here, but I am not convinced that the construal of events is the all encompassing basis for sentence structure. What about sharing information about your inner state, what you need, what you want, what you feel towards something, requesting favors from others, telling others what to do, etc?
Reply: Again, this relates to attitudes. Section 4.1 and further additions hopefully addresses this issue.
Reviewer 3 Report
This paper offers an overview of the author's previous research at the intersection of semantics and evolutionary thinking.
The main claim is that human language is organized around "event cognition", and it suggests that event cognition evolved for planning.
It is not totally clear to me whether the claim is that event cognition and collective planning through communication co-evolved. The paper is nevertheless very interesting to read. Though I sympathesize with most of the ideas presented here, I wonder whether the way they are presented here is fully appropriate for 'Entropy'.
The style of the paper reminds a tutorial. Instead of submitting ideas to the readers' critical wisdom and arguing about them, the author makes many assertions supported by his previous publications. Eventually, we wonder what is really new to this paper, as compared with those previous writings.
Though aspects of the paper sound like modelling (esp. the force-vector model), there's nothing in the paper that is close to the kind of model I thought 'Entropy' was expecting.
More specific points:
- Two notions are ambiguous to me.
1. The notion of causation (e.g. line 287). The author seems to use it to encompass two different things: the agent role (which causes some effect), and explanatory causes (found through abduction).
2. The notion of event. In normal English, events refer to "who did what to whom somewhere at some specific date". When the author says: "My proposal is that sentences refer to events." (544), the author's statement obviously doesn't apply to his own sentence if 'event' has its normal meaning (see also note 7). So he is considering another sense of 'event' (514). This should be made clear. The problem is that it seems to weaken the evolutionary claim.
- Talking about hunting does not bring us any closer to an evolutionary claim. In particular, taking example from contemporary hunters is meaningless. Those people have the same cognition as you and me. So you should take your examples from what you master (e.g. advising a PhD student or writing a Python instruction) rather than imagining putative cognitive processes going on in hunters' minds when using poison. (lines 249, 314, 321, 357)
- line 238: "For such new behaviours the throwers needed to find a way to reason about the physical effects". The verb "need" reveals a lack of proper argumentation. Other explanations (e.g. correlation) can be imagined. One could overuse "need" to describe animal hunting, for instance, and reach absurd conclusions.
- the link between the cognitive argument and the evolutionary one remains loose. Knowing that "event cognition" and the ability to use it in communication is useful for planning (376, 417, 450, 679) does not indicate that "event cognition" was selected for that purpose (language is useful for poetry, but might not have been selected for that purpose). The link with gossip (501) could be more convincing, but it is merely mentioned.
To sum up: I would have preferred a more concise description of "event cognition" and of its evidence in language, followed by a clear argument about its role in language evolution: why did it evolve in the first place, and what are the expected consequences. I suppose that this is what the author wanted to do, but the line of argument is too much diluted for it to be clear enough (to my taste).
Minor points:
The contribution of Len Talmy to the force-vector model is mentioned, but maybe not sufficiently acknowledged.
Author Response
This paper offers an overview of the author's previous research at the intersection of semantics and evolutionary thinking.
The main claim is that human language is organized around "event cognition", and it suggests that event cognition evolved for planning.
It is not totally clear to me whether the claim is that event cognition and collective planning through communication co-evolved.
Reply: The claim is rather that event cognition evolved as a consequence of the extended causal cognition and that this opened up for more advanced collective planning. I have made this clearer in the text.
The paper is nevertheless very interesting to read. Though I sympathesize with most of the ideas presented here, I wonder whether the way they are presented here is fully appropriate for 'Entropy'.
Reply: The article is written for the special issue “Complexity and evolution”. I have also added a paragraph concerning the relation between causal generics and information at the end of section 3.1
The style of the paper reminds a tutorial. Instead of submitting ideas to the readers' critical wisdom and arguing about them, the author makes many assertions supported by his previous publications. Eventually, we wonder what is really new to this paper, as compared with those previous writings.
Reply: I take this comment as a positive appraisal. It is my intention to put together pieces from my earlier work (and that of others) into a coherent and convincing narrative. As far as I know, this is the first article that shows the importance of causal thinking for the evolution of language. If the story holds water, this will generate a greater unification of different theories of the evolution of cognition. Since the article covers several fields, it is necessary to write it in the style of a tutorial.
Though aspects of the paper sound like modelling (esp. the force-vector model), there's nothing in the paper that is close to the kind of model I thought 'Entropy' was expecting.
Reply: I agree that a formal model would make the paper stronger, but given its wide scope it is difficult to present such a model in detail. More formal details of the model can be found in Gärdenfors, P. and Warglien, M. (2012), Warglien, M., Gärdenfors, P. & Westera, M. (2012), and Gärdenfors, P., Jost, J., & Warglien, M. (2018). I have added pointers to these articles.
More specific points:
- Two notions are ambiguous to me.
1. The notion of causation (e.g. line 287). The author seems to use it to encompass two different things: the agent role (which causes some effect), and explanatory causes (found through abduction).
Reply: This is a good point. I have added a paragraph here describing the distinction.
- The notion of event. In normal English, events refer to "who did what to whom somewhere at some specific date". When the author says: "My proposal is that sentences refer to events." (544), the author's statement obviously doesn't apply to his own sentence if 'event' has its normal meaning (see also note 7). So he is considering another sense of 'event' (514). This should be made clear. The problem is that it seems to weaken the evolutionary claim.
Reply: This is a valid point. I have added a new subsection 4.1 describing different kinds of communication and in subsection 5.1 the thesis that sentences refer to events is constrained to a subclass of declarative sentences. There I also discuss other types of sentences.
- Talking about hunting does not bring us any closer to an evolutionary claim. In particular, taking example from contemporary hunters is meaningless. Those people have the same cognition as you and me. So you should take your examples from what you master (e.g. advising a PhD student or writing a Python instruction) rather than imagining putative cognitive processes going on in hunters' minds when using poison. (lines 249, 314, 321, 357)
Reply: I agree that many hunting examples have been used in many just-so evolutionary stories. It is also true that I do not master hunting procedures. However, the first example comes from my collaborator Marlize Lombard (Bradfield et al 2015, Gärdenfors & Lombard 2020) who is a leading expert on early hunting techniques. I agree that taking example from contemporary hunters is problematic. However, the example involving San hunters is not used to make an evolutionary point, but just to illustrate that planning involves event cognition. Nor is the example on line 257 an evolutionary one.
- line 238: "For such new behaviours the throwers needed to find a way to reason about the physical effects". The verb "need" reveals a lack of proper argumentation. Other explanations (e.g. correlation) can be imagined. One could overuse "need" to describe animal hunting, for instance, and reach absurd conclusions.
Reply: Valid point. The sentence has been rewritten.
- the link between the cognitive argument and the evolutionary one remains loose. Knowing that "event cognition" and the ability to use it in communication is useful for planning (376, 417, 450, 679) does not indicate that "event cognition" was selected for that purpose (language is useful for poetry, but might not have been selected for that purpose). The link with gossip (501) could be more convincing, but it is merely mentioned.
Reply: Valid point, but I am not claiming that event cognition was selected for the purpose of improving planning, only that once event cognition is in place, it opens up for forms of communication that allow more advanced planning.
To sum up: I would have preferred a more concise description of "event cognition" and of its evidence in language, followed by a clear argument about its role in language evolution: why did it evolve in the first place, and what are the expected consequences. I suppose that this is what the author wanted to do, but the line of argument is too much diluted for it to be clear enough (to my taste).
Reply: I agree that the argumentation is somewhat shallow, but since it covers connections between three large areas – causal reasoning, event cognition, and language – that have never been put together previously, this is unavoidable, lest the text becomes a book.
Minor points:
The contribution of Len Talmy to the force-vector model is mentioned, but maybe not sufficiently acknowledged.
Reply: I have added that he was a pioneer. A limitation of ‘Talmy’s force dynamic model is that it only applies to events where both forces and counterforces are involved. The force-vector model presented here is more general (and it is grounded in the theory of conceptual spaces)
Reviewer 4 Report
This article proposes evolutional justifications for the formation of the sentence structure in language, in particular relating it to the causal thinking ability of the human species, and the cognitive modeling of events. It is a well-written article with ideas and hypotheses well-justified with proper references. The topics are of interest: The article offers well-argued hypotheses to interesting questions such as why language is organized in sentences, and why word classes such as nouns and verbs emerge consistently over various languages. I recommend this article for publication.
The article naturally is highly hypothetical, as the main reasoning for such emergent properties of language are proposed to be related to the evolution of human cognition. However I do find the references well-used and justifying the ideas in general.
Some specific comments are:
- I cannot gather enough evidence from the article's references that infants have a "special cognitive system of representing forces". Is this point critical to the argumentation of the paper? I believe the author is making a connection from such an "automatic" force representation to a somehow automatic event representation, and this is where the points connect. However I cannot directly see this connection, and I am unsure why an automatic event representation somehow derives from an automatic force representation.
Furthermore, if I understand correctly, at some point that the author connects physical forces to "non-physical" forces, such as non-physical actions of an agent, however, is there any evidence for such a generalization? This part feels more speculative to me compared to the rest of the article. Does the author mean that the cognitive mechanism for the physical representation of the world is in later cognitive stages being reused as the cognitive mechanism for representing non-physical actions? As far as I understand this is the claim, and it furthermore depends on the hypothesis of the representation of forces being somehow "atomic", maybe "first-order" in the cognition.
I somehow fail to see how critical these claims are to the main claims of the paper, and if very critical, then I would recommend more detailed results to be cited from the references, or maybe clearer connections to be made in this part.
- About line 268, I was thinking about agentless events. First of all, most of these example events (falling, drowning, dying, growing) seem like events with agents to me, or maybe I misunderstand the meaning of agentless here. I would suggest maybe state-related events, eg. as in "It is cold", can be considered as agentless, but for example do not we mean the environment as the "agent" of the event here? I was rather curious of an agentless-event idea and I wasn't persuaded by the short discussion here, although I do understand that this is not the author's intention to discuss this, but I do find this somehow speculative.
- About line 287, "In philosophy and psychology, the causal relation between the action and the effect is typically analysed as holding between two events", I similarly fail to see how this can be the case/view point of these disciplines, though of course this can be related to my missing background knowledge. How are "two different events" necessary for a causal relation representation? Am I misunderstanding the expression here?
- About line 418, it feels speculative to assume that animals do not "communicate" while for example cooperative hunting. I rather would assume they communicate although not via language, but for example via intention-gestures or sounds that signals the others of their internal plans. Humans use these kind of intention-gestures all the time, for example turning towards the door to signal that they need to be leaving the room. So while animals may not be communicating their intentions, I find this claim highly speculative and not supported. I need to add that does not invalidate the point that the author is trying to make, since the author's main discussion discussion is rather that their plans are related to the "current physical context". I just think it may be too strong to further claim a zero-communication case.
- I was missing a clear definition for the construal, at least it was difficult for me to glean its intended meaning from the text. Line 558 gives it as "what is expressed in the sentence", but this feels very vague to me.
- In line 585, although I am not an expert, I was sceptical about nouns and verbs being the only stable word classes cross-linguistically. Aren't other classes such as prepositions also exist over various languages, albeit maybe in the form of suffixes and prefixes? With my very limited knowledge, I would assume languages around the world could more or less express similar meanings, which need way more than the agents and actions only. Certainly the location of an event is also a critical piece of information which needs to be conveyed somehow, and it is difficult to imagine a language that is incapable of doing so. Does the author mean that then nouns and verbs (existing as separate "tokens") first-order elements of a language, whereas suffixes and prefixes are not first-order elements? However, if the meaning can as well be given through these, does it matter if they are separate tokens or not?
Minor comments:
- Line 95: "Sentences for... " > Malformed sentence
- Line 96: auses > causes
- Line 433: 2010:6 > 2010?
Author Response
This article proposes evolutional justifications for the formation of the sentence structure in language, in particular relating it to the causal thinking ability of the human species, and the cognitive modeling of events. It is a well-written article with ideas and hypotheses well-justified with proper references. The topics are of interest: The article offers well-argued hypotheses to interesting questions such as why language is organized in sentences, and why word classes such as nouns and verbs emerge consistently over various languages. I recommend this article for publication.
The article naturally is highly hypothetical, as the main reasoning for such emergent properties of language are proposed to be related to the evolution of human cognition. However I do find the references well-used and justifying the ideas in general.
Some specific comments are:
- I cannot gather enough evidence from the article's references that infants have a "special cognitive system of representing forces". Is this point critical to the argumentation of the paper? I believe the author is making a connection from such an "automatic" force representation to a somehow automatic event representation, and this is where the points connect. However I cannot directly see this connection, and I am unsure why an automatic event representation somehow derives from an automatic force representation.
Reply: This is Leslie’s (1995) conclusion and it is not critical to the paper. Wolff’s extensive series of experiments provides strong evidence for ‘force thinking’ in adults, which is more important for my argument. Non-human animals can reason about event involving actions – their own or those of others. The important step in reaching the more general event representation presented in the paper comes when not only actions but also other types of forces can function as causes in events. I have added this at the end of subsection 2.1.
Furthermore, if I understand correctly, at some point that the author connects physical forces to "non-physical" forces, such as non-physical actions of an agent, however, is there any evidence for such a generalization? This part feels more speculative to me compared to the rest of the article. Does the author mean that the cognitive mechanism for the physical representation of the world is in later cognitive stages being reused as the cognitive mechanism for representing non-physical actions? As far as I understand this is the claim, and it furthermore depends on the hypothesis of the representation of forces being somehow "atomic", maybe "first-order" in the cognition.
Reply: The proposal is that, among other things, humans see components of a theory of mind such as intentions, desires and beliefs as “mental forces” that act as causes generating behavioural effects. This proposal is elaborated in Lombard and Gärdenfors (2021) where also various types of evidence are presented. (Further support are also all the force metaphors we use when we talk about interpersonal relations.)
I somehow fail to see how critical these claims are to the main claims of the paper, and if very critical, then I would recommend more detailed results to be cited from the references, or maybe clearer connections to be made in this part.
Reply: They are not critical to the main claims, but – in another article – their consequences for language could be elaborated. The roles of intentions, desires and beliefs strongly influence how and about what we communicate.
- About line 268, I was thinking about agentless events. First of all, most of these example events (falling, drowning, dying, growing) seem like events with agents to me, or maybe I misunderstand the meaning of agentless here. I would suggest maybe state-related events, eg. as in "It is cold", can be considered as agentless, but for example do not we mean the environment as the "agent" of the event here? I was rather curious of an agentless-event idea and I wasn't persuaded by the short discussion here, although I do understand that this is not the author's intention to discuss this, but I do find this somehow speculative.
Reply: This is an interesting point, but somewhat tangential to the main argument. Grammaticalization has lead Indoeuropean language to demand a subject as in “It is cold” or “It is raining”. This may be the reason that we want to look for an agent such as “the environment” or “gravitation”. But in many other languages, subjects are not required. (Even in Italian “it is raining” is expressed by just a verb “piove”).
- About line 287, "In philosophy and psychology, the causal relation between the action and the effect is typically analysed as holding between two events", I similarly fail to see how this can be the case/view point of these disciplines, though of course this can be related to my missing background knowledge. How are "two different events" necessary for a causal relation representation? Am I misunderstanding the expression here?
Reply: Since Hume, this has indeed been the tradition in philosophy. The reason is perhaps that the cause, e.g. the stone was thrown, can be expressed by one sentence, and the effect, e.g. “the window broke”, by another. As should be clear, I think that this division is a mistake and that a single event needs both components. However, an analysis of the philosophical background is beyond the scope of the article.
- About line 418, it feels speculative to assume that animals do not "communicate" while for example cooperative hunting. I rather would assume they communicate although not via language, but for example via intention-gestures or sounds that signals the others of their internal plans. Humans use these kind of intention-gestures all the time, for example turning towards the door to signal that they need to be leaving the room. So while animals may not be communicating their intentions, I find this claim highly speculative and not supported. I need to add that does not invalidate the point that the author is trying to make, since the author's main discussion discussion is rather that their plans are related to the "current physical context". I just think it may be too strong to further claim a zero-communication case.
Reply: Yes and no. What I write is that animals do not communicate their plans. Of course, animals are reading the behaviour and maybe even the intentions of other individuals and use that to determine their own behaviour. This can be seen as a form of communication that improves the cooperation. The question is whether this kind of communication is intentional. Also when a human is turning to the door and the onlooker infers that the person will be leaving, this is non-intentional communication. I have made it clear that I refer to intentional communication.
- I was missing a clear definition for the construal, at least it was difficult for me to glean its intended meaning from the text. Line 558 gives it as "what is expressed in the sentence", but this feels very vague to me.
Reply: A definition has been added.
- In line 585, although I am not an expert, I was sceptical about nouns and verbs being the only stable word classes cross-linguistically. Aren't other classes such as prepositions also exist over various languages, albeit maybe in the form of suffixes and prefixes? With my very limited knowledge, I would assume languages around the world could more or less express similar meanings, which need way more than the agents and actions only. Certainly the location of an event is also a critical piece of information which needs to be conveyed somehow, and it is difficult to imagine a language that is incapable of doing so. Does the author mean that then nouns and verbs (existing as separate "tokens") first-order elements of a language, whereas suffixes and prefixes are not first-order elements? However, if the meaning can as well be given through these, does it matter if they are separate tokens or not?
Reply: Here I am following Heine and Kuteva’s (2007) analysis which is very thorough. Locations are expressed by nouns (or names) and much of the work that is done by prepositions can be handled by merely using location nouns. Prepositions basically express relations between nouns (not just spatial but also force relations – see Gärdenfors 2014).
Minor comments:
- Line 95: "Sentences for... " > Malformed sentence
- Line 96: auses > causes
Reply: Corrected
- Line 433: 2010:6 > 2010?
Reply: This is a reference to the page number.
Round 2
Reviewer 2 Report
Thank you for the clarifying revisions.
Reviewer 3 Report
I'm OK with the new version and the author's response.